# Differential Hebbian learning with time-continuous signals for active noise reduction

**Konstantin Möller[1]☉, David Kappel[1]☉¤, Minija Tamosiunaite[1,2], Christian Tetzlaff[3], Bernd Porr[4], Florentin Wörgötter[1]\***

**1** Third Institute of Physics and Bernstein Center for Computational Neuroscience, Univ. Göttingen, Göttingen, Germany, **2** Faculty of Informatics, Vytautas Magnus University, Kaunas, Lithuania, **3** Department of Computational Synaptic Physiology, Institute for Neuro- and Sensory Physiology, University Medical Center Göttingen, Georg-August University, Göttingen, Germany, **4** Biomedical Engineering School of Engineering University of Glasgow, Glasgow, Scotland

☉ These authors contributed equally to this work.
¤ Current address: Institut für Neuroinformatik, Ruhr-Universität Bochum, Bochum, Germany
* worgott@gwdg.de

**Data Availability Statement:** All relevant data are within the paper.

**Funding:** F.W. and C.T. received funding from the European Commission under H2020 grant

## Abstract

Spike timing-dependent plasticity, related to differential Hebb-rules, has become a leading paradigm in neuronal learning, because weights can grow or shrink depending on the timing of pre- and post-synaptic signals. Here we use this paradigm to reduce unwanted (acoustic) noise. Our system relies on heterosynaptic differential Hebbian learning and we show that it can efficiently eliminate noise by up to -140 dB in multi-microphone setups under various conditions. The system quickly learns, most often within a few seconds, and it is robust with respect to different geometrical microphone configurations, too. Hence, this theoretical study demonstrates that it is possible to successfully transfer differential Hebbian learning, derived from the neurosciences, into a technical domain.

## Introduction

Hebb rules [1] have been employed in a wide variety of (unsupervised) learning tasks and exist in many versions. The literature about this topic is vast and extends from its origins in the neurosciences into many theoretical, but also application-driven, contributions in artificial neural networks (see [2] for a short review). In this study, we focus on the use of Hebbian learning to address the problem of learning to suppress (acoustic) noise in time continuous signals. We will show that the methods introduced here, while derived from neuronal models of plasticity, can be successfully transferred into this technical domain, too.

The background for this is the fact that Hebbian plasticity can rely on the temporal signal sequence. This had first been discovered in 1997 (spike timing dependent plasticity [3, 4]), where the sequence of pre- and post-synaptic signals determines whether a synaptic weight will grow (Long Term Potentiation, LTP) or shrink (Long Term Depression, LTD). Theoreticians had been intrigued by this finding, because learning rules that allow for both, weight growth and shrinkage, may lead to better stability in a neural network. Accordingly also here many models had been designed and tested mainly until about 2010 (see [5] for a review).

agreement 899265, FET-Open Project "ADOPD". https://ec.europa.eu/info/research-and-innovation/funding/funding-opportunities/funding-programmes-and-open-calls/horizon-2020_en The funders had no role in study design, data collection and analysis, decision to publish, or preparation of the manuscript.

**Competing interests:** The authors have declared that no competing interests exist.

Spike timing dependent plasticity can be linked to differential Hebbian learning given by $\frac{d\omega}{dt} = \mu u \frac{dv}{dt}$, with $\omega$ a synaptic weight, $0 < \mu \ll 1$ a learning rate, $u$ the input, and $v$ the output of a neuron [6] with the advantage that differential Hebbian learning allows treating these systems in a closed form.

These types of learning rules have been especially useful, however, when focusing on the problem of temporal sequence learning [7]. In this case at least two (but often more) input signals $x_0, x_1, \ldots, x_n$ exist, which are usually some *events* that are correlated to each other, but with certain delays between them. For two signals $x_0, x_1$, this system is related to classical and/or instrumental conditioning and had been modeled around 1980 by using correlation-based stimulus-substitution models [8]. In real life this can happen, for example, when we first feel a heat pulse that precedes a stabbing pain event on touching a hot surface. In order to avoid such painful events, it is advantageous to learn reacting to the earlier event, not having to wait for the later one. The same is true for many sensor events where—for example—a visual signal may be predictive for a collision (touch), or an auditory signal for an approaching predator. In all these cases, it is better to learn reacting to the earlier event and not to the later (potentially dangerous) one.

With differential Hebbian learning rules introduced by us (isotropic sequence order learning, ISO [7]; and input correlation learning, ICO [9]), the agent can learn an anticipatory action to react early and avoid the late event. The late event $x_0$ acts as the reference and provides the error signal for the learning. Importantly, it can be shown that learning is converging with the vanishing of this error signal, hence as soon as $x_0 = 0$ [7, 9]. Thus, as soon as the later signal is successfully avoided learning stops.

Here we make use of this property to eliminate (to "avoid") noise at a local microphone $x_0$ learning to correctly compensate for it using a set of (predictive) distant microphones $x_1, \ldots, x_n$, where learning will stop as soon as the local microphone $x_0$ does not hear noise any longer ($x_0 = 0$).

## State of the art

The most popular noise cancellation algorithm by far is the least mean square (LMS) algorithm [10–13], which outperforms conventional filters [14], optimal stationary linear filters [15], or adaptive smoothing filters [16]. The general idea here is to adaptively cancel out noise with the help of an opposing signal so that the result is noise free. This requires a noise reference, which, if appropriately filtered, generates the opposing signal. Fig 1 shows the general principle, where noise $x_1, x_2, x_3$ is sent through an adaptive filter and then its output $y(n)$ eliminates the noise contained in $d(n)$ at the summation point $\Sigma$. The signal $x_0$ is both, an error signal and the output of the noise canceller. If the elimination has not been perfect, the resulting error can be used to tune the coefficients of the filter, for example, by using the delta rule [17,

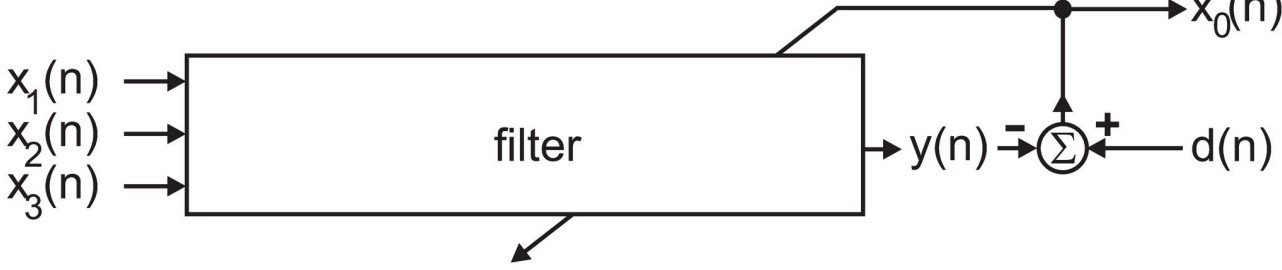

**Fig 1. Standard circuit for LMS-based noise cancellation.**

18]. While Widrow et al. [10] only addressed noise cancellation by digital subtraction, Elliot and Nelson [19] discuss the elimination of noise by mixing acoustically opposing sound waves [20]. This is nowadays widely used in noise cancelling headsets [21]. Traditional noise cancelling headsets employ a real error microphone, but also a simulated error microphones can be used, which, in turn, tunes the LMS filter [18]. Recently, the concept of an acoustical error has been translated to a setup with multiple error microphones and speakers to eliminate noise in a seminar room, which is still based on the LMS algorithm but combined with Eigen analysis to tackle the multiple cross correlations between different error microphone signals and speakers outputs [22].

However, at the heart of both digital or acoustic cancellation is the delta rule [17], which changes the filter coefficients by correlating the error signal with the reference noise. The delta rule, as is the case with the Hebbian learning rule, is symmetric in time. In contrast ICO learning uses the derivative of the error signal, which is a predictor of the error signal because of its phase lead.

We will show that our system, based on the ICO rule, will achieve noise reduction in a simulation to a high degree, competitive with existing standard noise reduction methods. In addition to the experimental results, we provide at then end the analytical solution for the weight growths under the assumption of a stationary power spectrum of the incoming signals.

## Materials and methods

### Active noise reduction: General setup

We consider the active noise reduction problem outlined in Fig 2A. The goal here is to eliminate background noise by applying a suitable anti noise signal that silences the acoustic environment. To do so an array of *control microphones* $x_1$, $x_2$ and $x_3$, is used to record ambient noise sources. We consider here a simple simulation of the noise reduction problem, where non-linear effects are ignored. A *reference microphone* $x_0$ is used at the recording site to drive the learning of the parameters for the anti noise signal. We tested different geometrical configurations for the relation between control- and reference microphones as shown in the Results section.

### Learning rule

The conventional ICO learning rule is given by (see Fig 2B):

$$\frac{d}{dt} w_i(t) = \mu x_i(t) x_0'(t) \tag{1}$$

where $x_i$ are the control inputs, $w_i$ their synaptic weights, $x_0$ the reference input and $\mu \ll 1$ a learning rate. Note that, ICO uses the derivative, annotated as $x_0'$, for learning, where we assume that $x_1$ represents an early signal which is correlated to the later occurring signal $x_0$.

The use of a derivative in the differential Hebbian rule lends itself to some intuition behind the mechanism of ICO learning. The derivative is a predictor of the signal's next moment's development. Hence, as soon as there is a correlation between an earlier and a later event the derivative will lead to an upregulation of the synapse that belongs to the earlier event until the neuron will respond to it reliably.

We employ here a slightly modified ICO learning rule, which uses the momentum (or moving exponential average) of the derivative of the reference signal $x_0(t)$ given by:

$$X(t) = \beta X(t-1) + (1-\beta) x_0'(t) \tag{2}$$

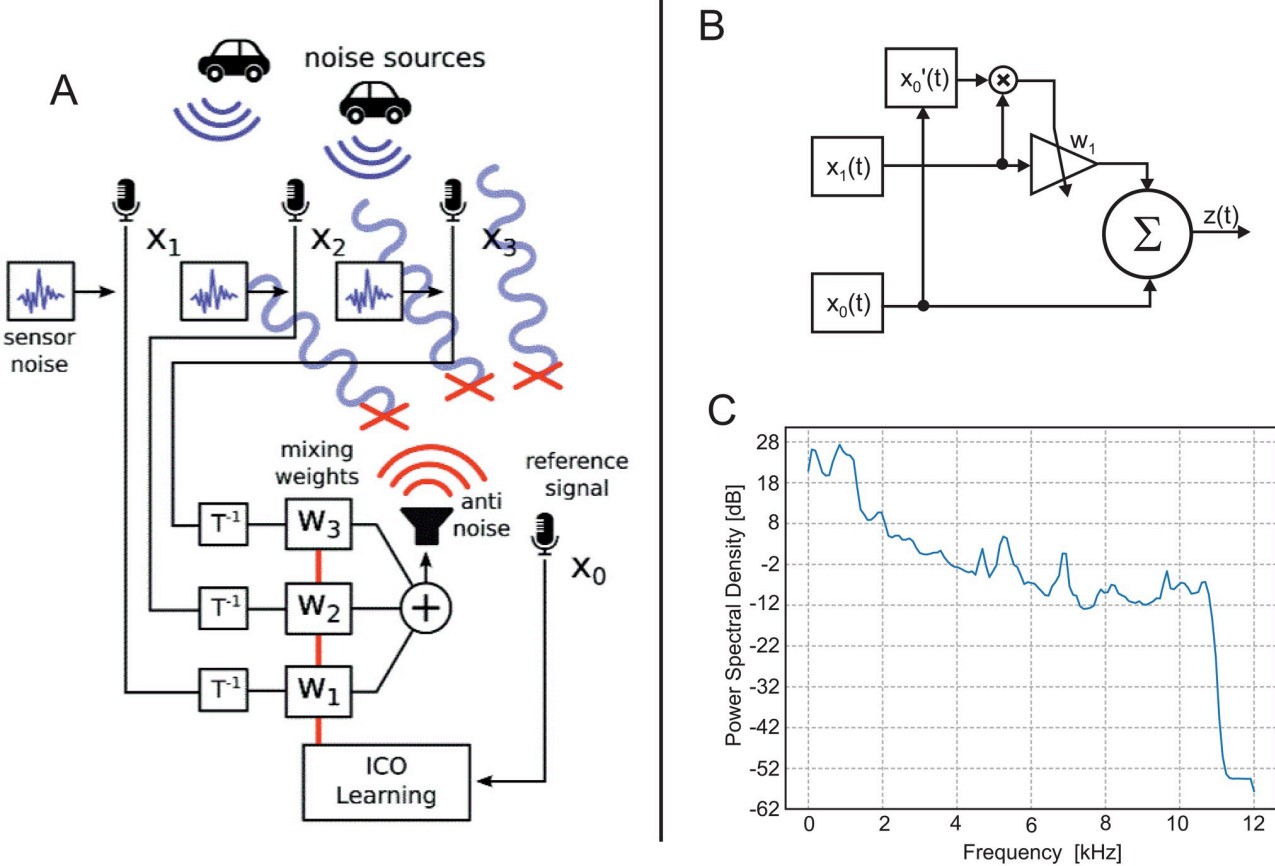

**Fig 2. ICO learning for active noise reduction.** A) Illustration of the active noise cancelling architecture. A set of control microphones, $x_1$, $x_2$ and $x_3$, records ambient noise. ICO learning is used to learn parameters for mixing the recorded noise to produce a suitable anti-noise to cancel at the reference recording site ($x_0$) that receives signals with a propagation delay $\Delta t$ relative to the earlier-arriving signals at the control microphones. B) Schematic of the conventional ICO rule. The triangle represents a synapse with changing weight. C) Power spectrum of the noise used for all tests.

where $\beta = 0.9$. The momentum is used to achieve smoother weight estimation and leads to better results than using the derivative of the late error signal $x_0$ directly.

Weights are updated using this learning rule

$$w'_i(t) = \mu x_i(t) X(t) \tag{3}$$

At simulation start weights $w_i$, $i > 0$ were set to 0, $x_0$ enters the summation node (Fig 2B) with a factor of one.

## Noise reduction mechanism

Based on the setup in Fig 2A and to solve the noise cancellation problem by generating an anti noise signal, an error signal of form

$$x_0(t) = x_0^*(t) - z(t) \tag{4}$$

is defined. Here $x_0^*(t)$ is the resulting noise signal at $x_0$, hence, the sum of all noise sources at the reference microphone. The function $z(t)$ is the generated anti noise signal given by:

$$z(t) = \sum w_i x_i(t - \Delta t), \tag{5}$$

where $\Delta t$ is the sound travel time delay between early and late inputs. The goal of the system is to achieve $x_0(t) = 0 \ \forall t > t_0$, by learning appropriate values for $w_i$, $i > 0$. The intuition behind this is that ICO learning is supposed to adjust the weights of the compensatory signal $z(t)$ such that indeed there is no noise audible at the reference microphone $x_0$ any longer.

## Technical simulation details

The general setup had been described above. Here we add some technical details, which should allow reproducing all results. The speed of sound is assumed to be 343 *m/s*. All noise sources are more than one meter away from microphones and in general we assume for all experiments a short distance from the sound source(s) to the different microphones on the order of a few meters (see Results). Thus, in general, the amplitude of the audio signals is multiplied by $1/d$, where $d$ is the distance from the source traveled in meters. This relation is also used to address effects of distance-dependent signal attenuation when considering larger distances between sound source and recording microphones (see Fig 3F).

For the ambient noise sources we used freely available recordings from https://freesound. org/ (creative commons license). If the audio file has two channels (stereo), we have down-mixed it to mono by taking the average of both channels. The audio files for the noise source were sampled at 22 *kHz*. The simulation, however, runs with 24000 samples per second, leading to a frequency cut-off at 12 *kHz*, which is above the cut-off frequency of the noise shown by the power spectrum in Fig 2C. We had tested the system with several noise sound files from the above repository but results did not change in any substantial way and, therefore, we show results only from one file.

Different microphone configurations are tested as shown in the results figures below. Sound file inputs from the microphones were combined with mixing weights $\theta_i$, $i > 0$ taken randomly from the interval [0, 1] and then kept constant for a given simulation. In some cases a virtual "shielding" is employed to test interference between microphones. This is implemented by reducing the signal strength by 95% in all shielded microphones. See below for details. It is assumed, that all microphones are located between the sources and the reference microphone. The value of $\Delta t$ (sound travel time) in the update rule is determined by taking $\min(0, idx)$, where $idx$ is the time delay between the closest source and the reference (that source most likely also accounts for the largest part in the sum of signals of that microphone). If no source is closer than the reference microphone we set $\Delta t = 0$, to avoid acausal signals.

## Experiments and setups

Table 1 summarizes all experiments and their motivation for better navigation through the Results section below.

## Results and discussion

### Experiment 1 (Table 1)

Simulation results for the linear microphone arrangement from Fig 2A are shown in Fig 3. The geometrical arrangement is shown in the inset in panel B. Fig 3 shows that ICO learning is able to recover the parameters $w_i$, $i > 0$ after around 2 s learning time and weights stabilize as expected. Noise reduction settles at around -140 dB. This represents quite a high level, which can be appreciated from the following comparison: a jet plane noise at close distance is at about 140 dB, whispering speech or gentle wind in a forest results in approximately 20 dB.

Panel G addresses the important aspect how noise reduction changes when using larger distances for the microphone configuration. This is done by introducing a distance scaling factor

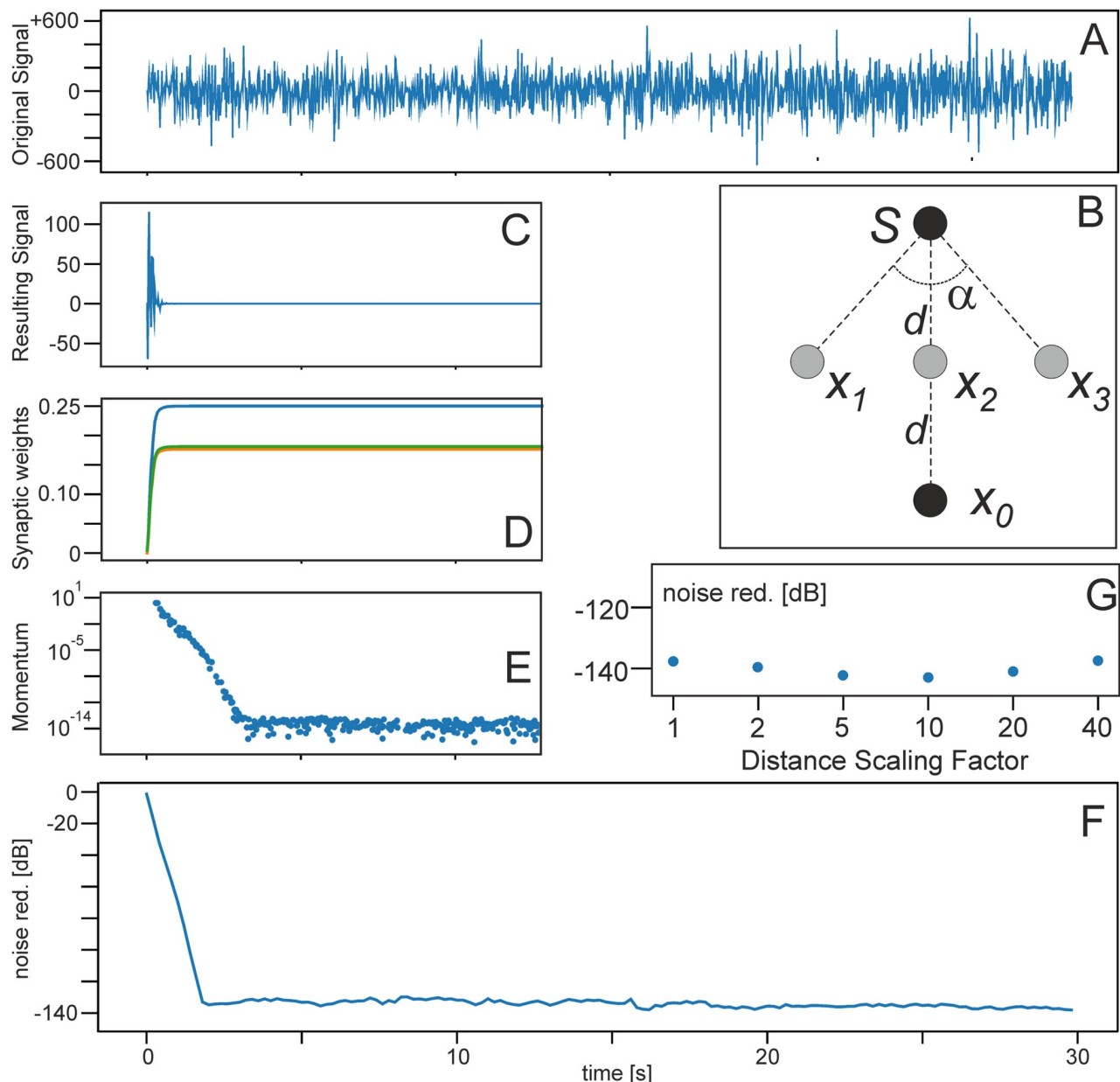

**Fig 3. Results for a linear microphone setup. A** Original mixed noise signal. **B** Microphone and noise-source geometry, $\alpha = 90°$, $d = 1.25\ m$. (C-E in arbitrary units and truncated after 12 $s$ as there are no more changes visible afterwards). Learning rate: $\mu = 1.0 \times 10^{-7}$. **C** effective noise at the reference site, **D** weight dynamics, green and orange curves are identical due to the symmetry of the microphone configuration and are here shifted a bit to make both visible, **E** development of the momentum, and **F** achieved noise reduction. **G** Noise reduction when scaling $d$ in the configuration with different factors.

and the results show that noise reduction remains the same. Note, however, that the learning rate needs to be increased for larger distances without which convergence will take much longer because the signal amplitude reduces with distance.

## Experiment 2 (Table 1)

So far we assumed perfect linear and stationary environment, that can be perfectly compensated using the ICO network. Here we turn to less ideal situations. In Fig 4 we investigate the

**Table 1. Experiments and goals.** Abbreviation 'mic.' stands for microphone.

| Experiment no. | Setup | Goal |
|---|---|---|
| 1) Linear setup (Fig 3) | Linear mic. setup with 90 deg. angle and different distances | Noise reduction for the most generic case at different distances |
| 2) Parameter drift (Fig 4) | Altering the control mics.' mixing parameters | Show speed of re-adjustment of the system to new parameters |
| 3) Different mic. configurations (Fig 5) | Various different mic. geometries | Show robustness against different configuration structures |
| 4) Shielding (Fig 6) | Mic. configurations with confused signal flow but shielding against this | Show effect of shielding against wrong sound signal flow |
| 5) Momentum (Fig 7) | Compare setup when using the momentum vs original ICO rule | Demonstrate improved efficiency when using the momentum |
| 6) Low pass (Fig 8) | Learning the correct filter characteristics of an unknown low pass | Show that the system can learn the transfer function of the environment |
| 7) Signal preservation (Fig 9) | Mixing a signal with noise | Show that signal is retained and noise eliminated |

impact of drifts in the parameters. After 10 seconds training time we changed the transfer function of the environment to different sound-mixing weights $\theta_1$, $\theta_2$ and $\theta_3$, which follow for every time step a drift given by $\theta_i - k \cdot (-1^i)$. As $i > 0$ is the weight-index, we get this way that $\theta_1$ increases per step by $k$, whereas $\theta_2$ decreases by $k$, etc., where different values for $k$ have been used. Fig 4 shows the results for a total duration of the parameter drift of 1 $s$. During the drift period performance deteriorates to some degree, but remains better than at the very beginning of the experiment. Then one can see that the system at the end of the drift period quickly relearns until a new stable set of parameters is reached. If the drift is small (small $k$) deterioration is less. Thus, in general ICO tracks parameter changes during the drift with a delay of a few hundred milliseconds. This delay causes some degradation of the noise reduction performance (Fig 4A) but performance stabilizes at a high level again after the drift.

## Experiment 3 (Table 1)

Fig 5 shows the achieved noise reduction for different microphone configurations at a distance of 2.5 $m$ between source and $x_0$. Even with a random distribution of 10 control microphones (panel D) one gets a substantial reduction, albeit—in this case—only after a long time. The other configurations (panel A-C) achieve this in only a few seconds. In C we have used either an initialization with zero weights (solid line) or with random weights, taken from the interval [0,1), (dashed line) for the control microphones. Random initialization performs slightly slower. Note that, in general, configurations with control microphones behind the reference microphone (in the direction of sound travel) perform less good than the others (e.g. panel c), which one would expect (see also Fig 6, next).

## Experiment 4 (Table 1)

If the roles of control- and reference microphones is purposefully confused for example by a symmetrical setup (inset in Fig 6), no noise reduction can be achieved (Fig 6A). However, when introducing a shielding of 95% (panel B) noise reduction is strong again. Note, shielding is placed against those noise sources, which are non-predictive, hence which are—viewed in the direction from a given $S$ to $x_0$—*behind* $x_0$. This works for different configurations in the same way. For example when using two opposing noise sources with shield the same fast noise reduction effect is observed as in Fig 3. See, e.g., Fig 7 for a more complex example.

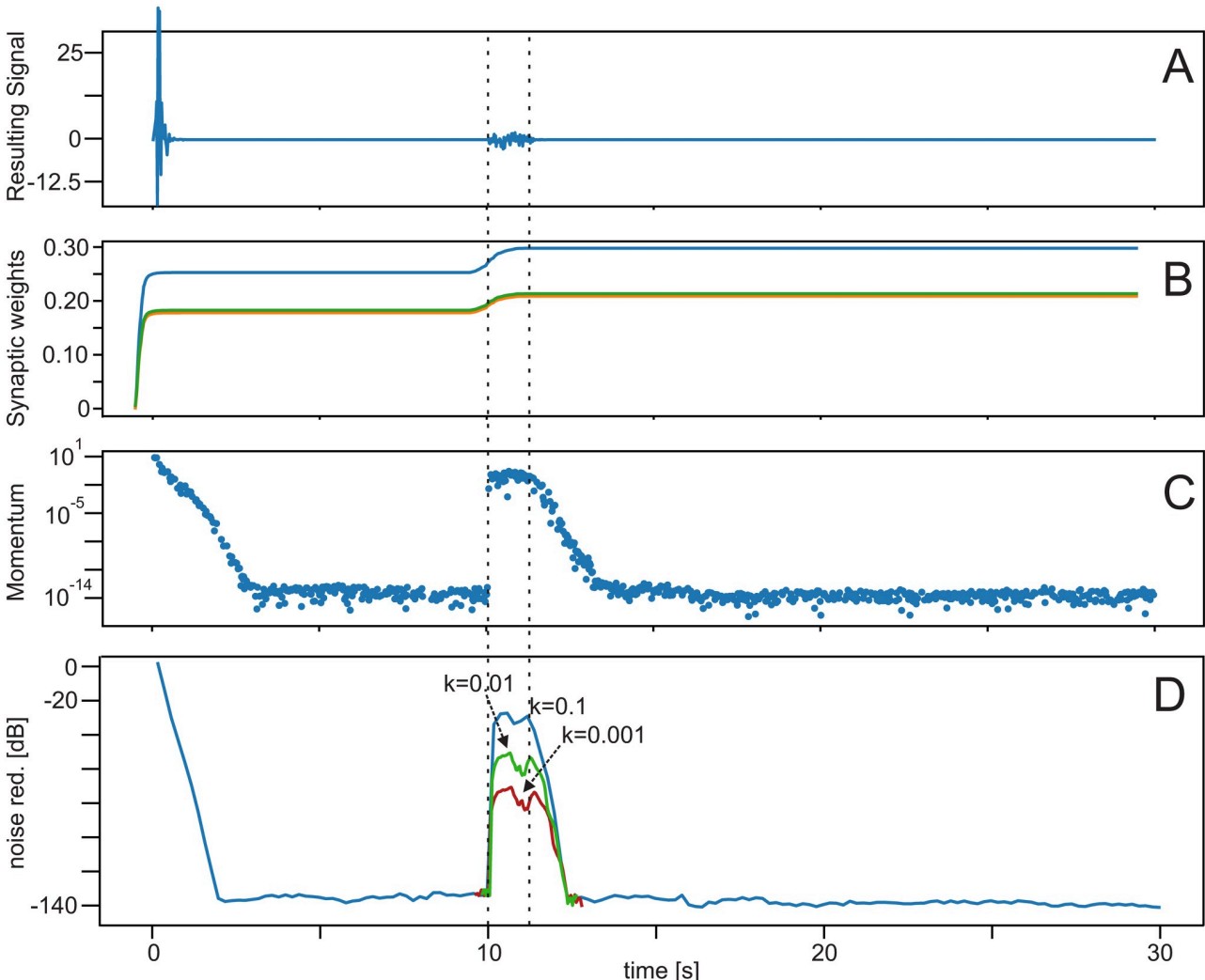

**Fig 4. Results for parameter drifts.** We used the linear microphone setup from Fig 3B. After 10 $s$ a parameter drift for 1 $s$ has been introduced (dashed lines). (A-C in arbitrary units). Learning rate: $\mu = 1.0 \times 10^{-7}$. **A** effective noise at the reference site. Before the parameter drift this curve is identical to Fig 3B, but here truncated in y-direction to make the effect of parameter drift visible. **B** weight dynamics, green and orange curves are identical due to the symmetry of the microphone configuration and are here shifted a bit to make both visible, **C** development of the momentum, and **D** achieved noise reduction for different drift rates $k$ (only parts of the red and green curves are shown).

### Experiment 5 (Table 1)

We use another configuration with multiple noise sources and shielding to demonstrate the efficiency of using the momentum instead of the original ICO rule (Fig 7). Panel A, without momentum, does not reach the final noise reduction level after 40 $s$ seen in panel B (with momentum) already after about 25 $s$.

### Experiment 6 (Table 1)

An interesting case concerns the aspect that signals at the reference microphone might have different frequency contents as those at the control microphones. For example, one could assume that signals further away are more strongly low-pass filtered as compared to signals from the same source closer by. In the example of car noise in Fig 2A, this might come from

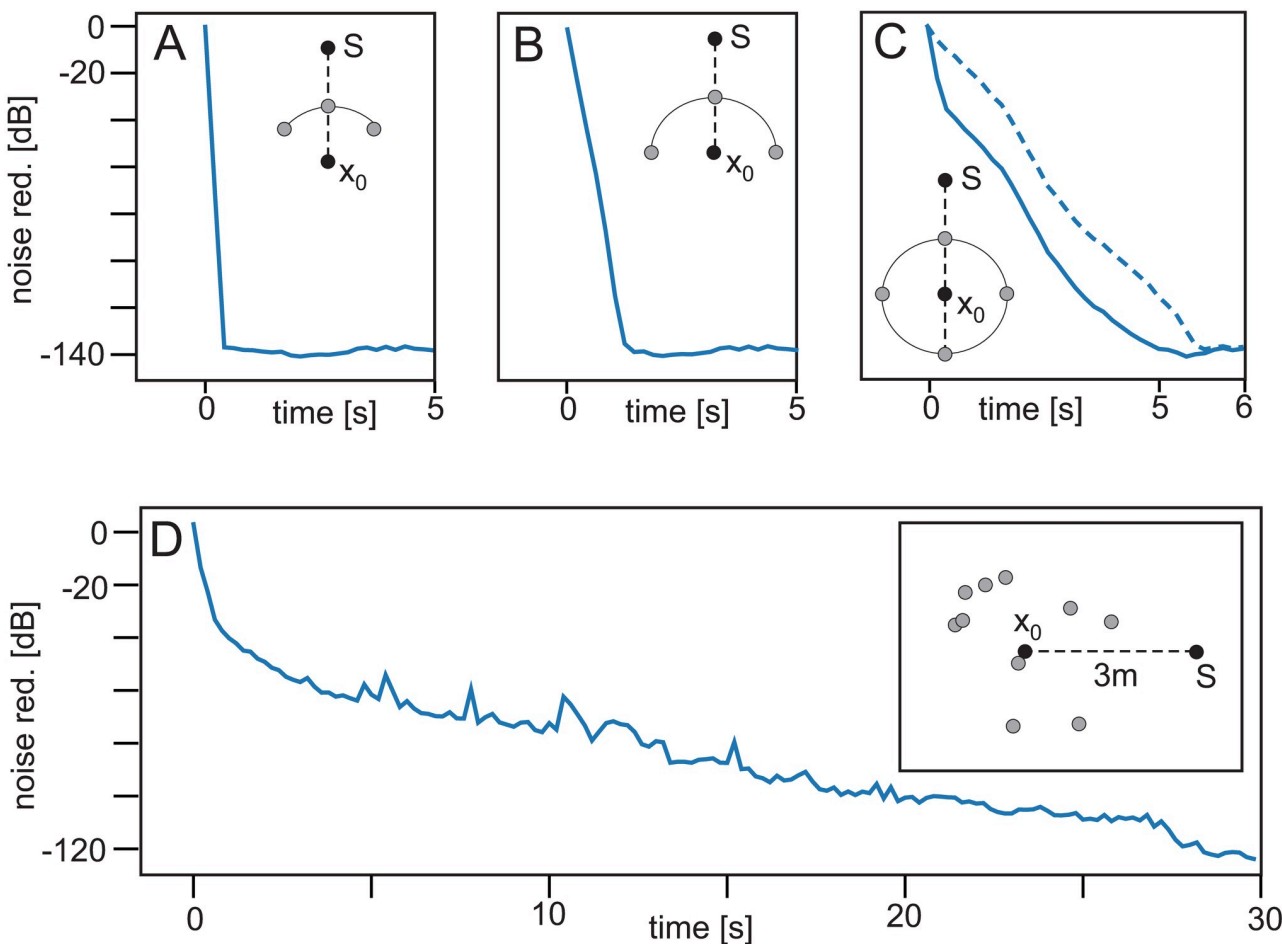

**Fig 5. Different microphone configurations.** Learning rate $\mu = 5 \times 10^{-7}$. **A** quarter circle, **B** half circle, A-B with 3 control microphones. **C** full circle with 4 control microphones, **D** random configuration with 10 control microphones.

some obstacles, like bushes, trees, etc. along the sound path. If this happens, as expected, noise reduction is completely lost and the result is almost identical to the untreated noise. Naively, one could try to employ some "world-knowledge" and try to explicitly model the signal path-way with *some* low-pass filter and insert this into the noise acquisition at the control micro-phones. However, if the filter cut-off frequency is wrong, noise reduction is severely impaired (Fig 8B), where we have used the same microphone configuration as in Fig 3. Here the cut-off frequency is 10 *kHz* at the reference microphone and 11 *kHz* at the three control microphones. A fifth order Butterworth filter has been used for the filtering. Both filters would have to have the same cut off to regain good performance. This could be achieved in principle by measuring the correct frequency cut-off, which—however—might be onerous. Instead, ICO learning can address this problem in an adaptive manner without much effort. In panel A we show the con-figuration, which can achieve this. The pathway to $x_0$ is filtered by the environment with a so-called "unknown" low-pass (here set to a cut-off of 10 *kHz*). Control microphone signals are split into three paths each with different low pass filters LP1, LP2, and LP3 with cut-offs of 9, 10 and 11 *kHz*. Panels C and D show that, after about 30 *s*, ICO learning has adjusted the weights of that reference microphone path with the correct filter (path 2) and we obtain a very high degree of noise reduction. Panel C shows that the weights will still change a bit until

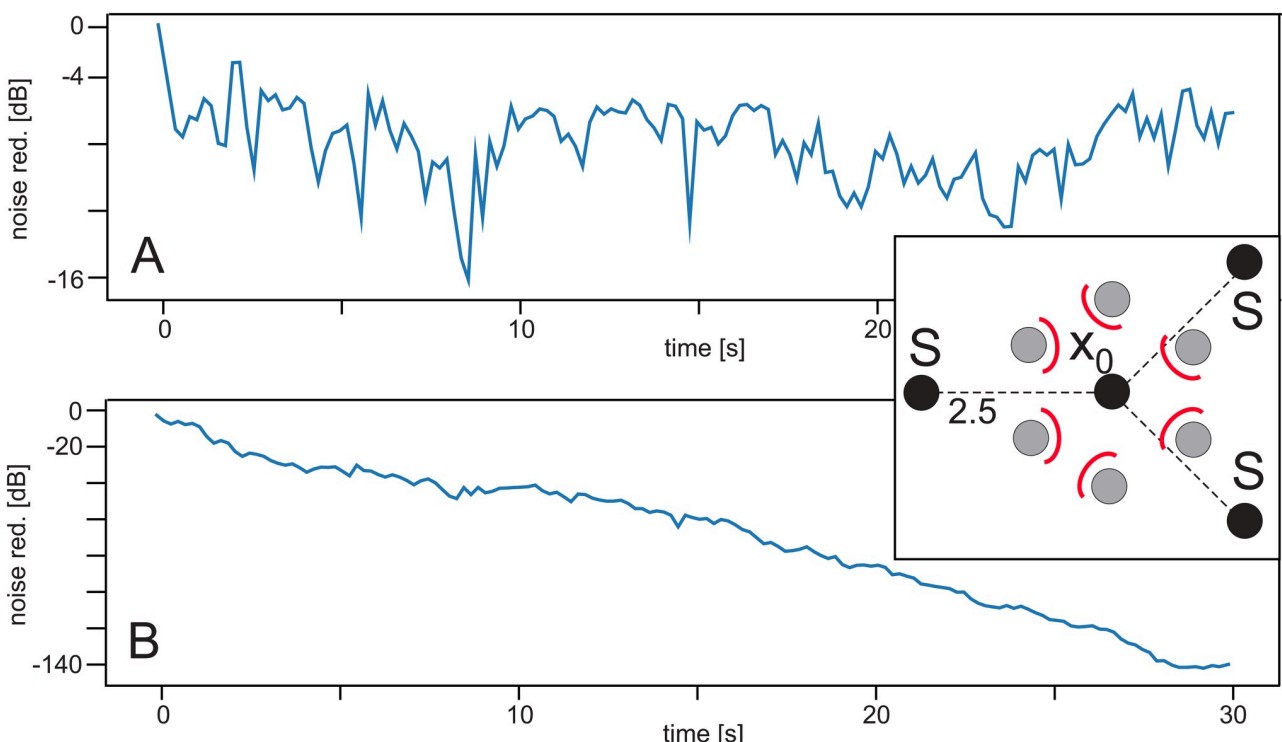

**Fig 6. Effect of shielding when using different noise sources.** Learning rate: $\mu = 7.5 \times 10^{-8}$. Configuration is shown as inset in panel B, small red arcs show the shields against the non-predictive noise sources. Distance from $S$ to $x_0$ was 2.5 $m$. **A** Noise reduction without shielding, **B** with shielding.

about 60 $s$, but noise reduction remained essentially the same after already 28 $s$. Note that only 6 of all 9 weights are visible as—due to the symmetry in the control microphone setup—some curves overlap. Clearly visible, however, is that, after an initial jump, only one microphone path increases in weights, while the others drop to zero. Furthermore note that in any technical application such a filter-finding process would have to be performed only once (or whenever the environment changes). Also note that in principle, many more paths with different filter characteristics could be used in case of greater insecurity about the "true" cut-off frequency of the unknown low pass from the environment.

### Experiment 7 (Table 1)

Fig 9 shows result for the situation when there is a relevant signal present at $x_0$, which should remain after filtering the noise. We have tested sine wave signals of different frequency up to 10kKz where—for visualization purposes—we show only some low frequency examples in the figure. The experiment was done in the following way. We use sine waves with different amplitudes $A_0$ and provided at the reference microphone the signal mix of $A_0 + noise$, whereas at the control microphones only the noise was present. The figure shows that the system removes the noise but leaves the signal intact. During the first 1-2 seconds, the system has not yet converged and one can see that the noise amplitude is as strong or even much stronger than the signal amplitude. After this time, ICO has eliminated the noise. In general, we obtained a noise reduction of >45 $dB$ for the complete sine-wave frequency range between DC and 10 $kHz$. The inset in A shows signal stability by the ratio of $A_0^{ICO}/A_0$, where $A_0^{ICO}$ is the amplitude of the

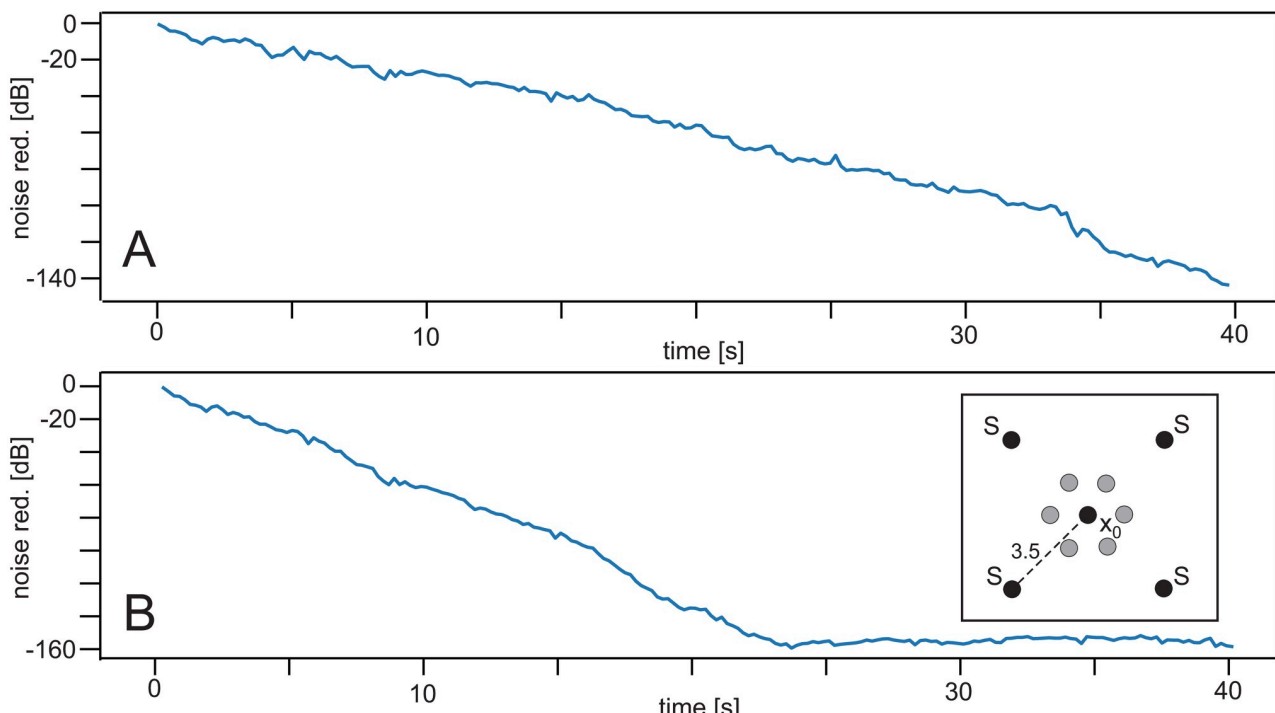

**Fig 7. Effect of using the momentum.** Learning rate: $\mu = 7.5 \times 10^{-8}$. Configuration is shown as inset in panel B. It contains a similar shielding as in Fig 6, which for clarity, is not drawn. Distance from S to $x_0$ was 3.5 *m*. **A** Using the original ICO rule, **B** Using the rule with momentum.

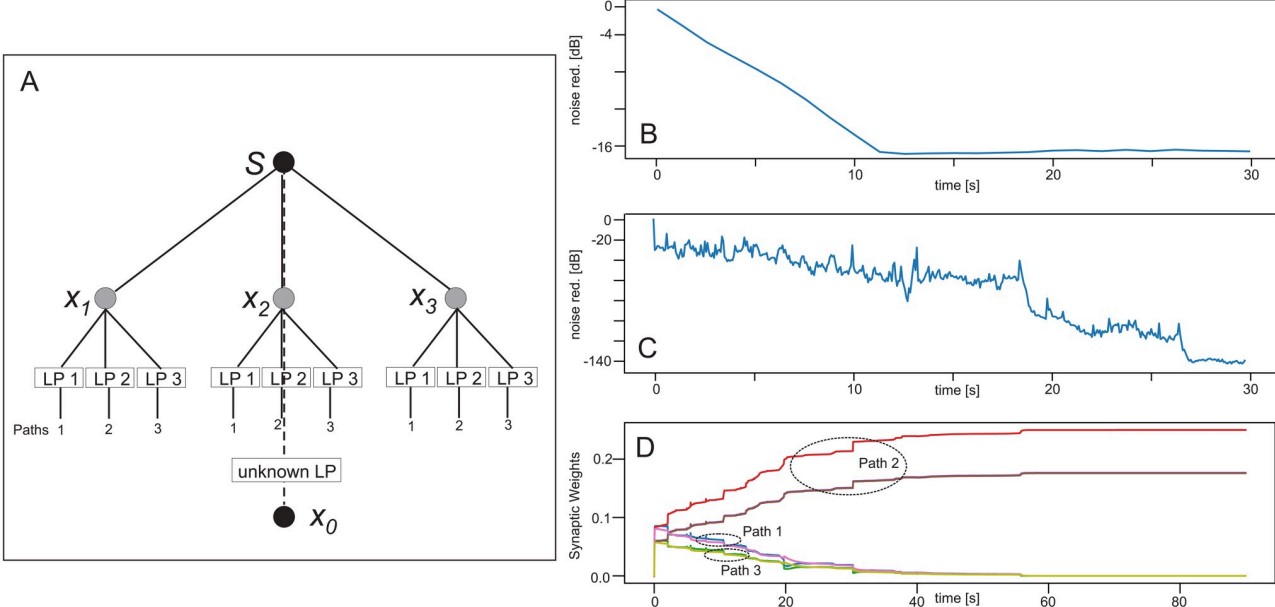

**Fig 8. Noise reduction when using low-pass filtered signals. A** Same geometrical configuration as in Fig 3 but now with split-up paths for the control microphones using different filters, where the cut-off for the reference microphone (so-called "unknown LP") was 10 *kHz*. Learning rate: $\mu = 1.0 \times 10^{-7}$. **B** Noise reduction for 3 conventional control microphones without path splitting, where only one low pass has been used with cut-off of 11 *kHz*. Note that the here-reached level was only about 16 dB. **C** Noise reduction for 3 paths from the control microphones. Path 1 with cut-off of 9 *kHz*, path 2 with 10 *kHz* and path 3 with 11 *kHz*. **D** Development of the synaptic weights of all paths. Path 2 shows weight growth and, thus, has been responsible for the resulting reduction of about 140 dB see in panel C.

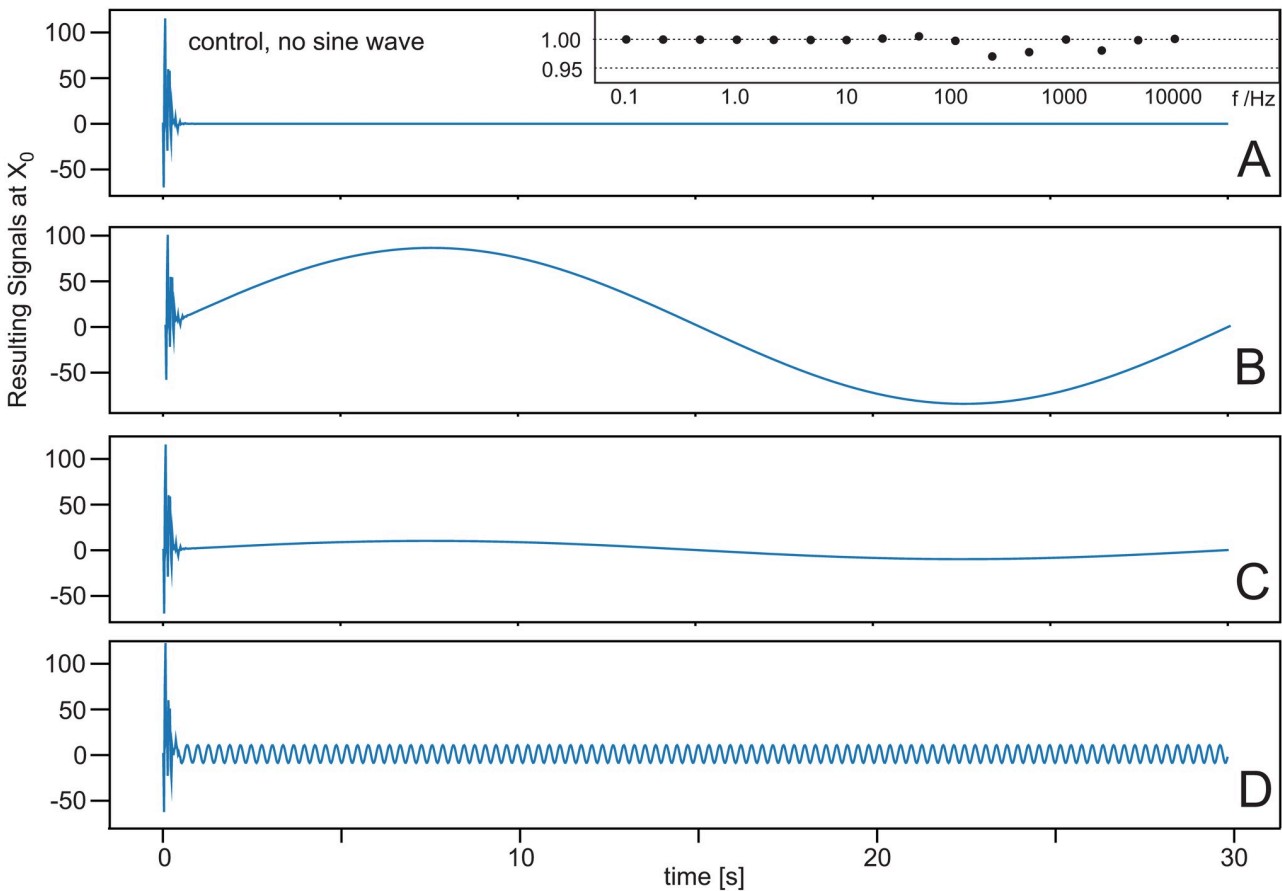

**Fig 9. Influence of noise cancellation on a sine-wave signal.** Geometrical configuration as in Fig 3. Learning rate: $\mu = 1.0 \times 10^{-7}$. **A** Control case without signal. **B** Sine wave with frequency 1/30 *Hz* and different amplitudes. **C** Same frequency as in B but amplitude reduced by a factor of 10. **D** Sine wave with frequency 100/30 *Hz*. The inset shows the ratio between the signal provided at $x_0$ without noise and the signal obtained after filtering the noise with ICO for a large range of frequencies.

sine wave after noise reduction. This ratio remains almost 1.0 across a large frequency range showing that the signal does not get destroyed by the ICO process.

## Analytical solution of the weight growth

Here we use the original ICO rule as, for this, we can provide some analytical insights, too. We consider time variable input functions $u_0(t)$ and $u_1(t)$ and the ICO learning rule $\frac{d}{dt}w(t) = \mu u_1(t)u'_0(t)$, where $u'_0(t)$ denotes the time derivative of $u_0(t)$ and with constant learning rate $\mu$. The dynamics of $w(t)$ can be solved directly in the time domain

$$\frac{d}{dt}w(t) = \mu u_1(t)u'_0(t) \quad \Leftrightarrow \quad w(t) = \mu \int_0^t u'_0(s)u_1(s)ds + w(0) \qquad (6)$$

To analyze the spectral behavior of the learning rule we assume that $u_0(t)$ and $u_1(t)$ are given by sine functions, with constant frequencies $\phi_0, \phi_1$ and constant phase shifts $\theta_0, \theta_1$.

$$u_0(t) = R_0 \sin(\phi_0 t + \theta_0)$$
$$u_1(t) = R_1 \sin(\phi_1 t + \theta_1)$$

where $R_0$ and $R_1$ are parameters to scale the amplitude of the sine functions. In general we can write:

$$Rsin(\phi t + \theta) = Bsin(\phi t) + Acos(\phi t)$$

with: $R^2 = A^2 + B^2$ and $\theta = arctan(B/A)$.

This way:

$$u'_0(t) = \phi_0(B_0cos(\phi_0) - A_0sin(\phi t))$$

We first treat the case where $\phi_0 \neq \phi_1$.

For the weight we get then:

$$w(t) = \mu \int u'_0(t)u_1(t)dt = \mu\phi_0 \int [B_0cos(\phi_0 t) - A_0sin(\phi_0 t)][A_1cos(\phi_1 t) + B_1sin(\phi_1 t)]dt \quad (7)$$

Hence:

$$w(t) = \mu\phi_0\left[\int B_0B_1cos(\phi_0 t)sin(\phi_1 t)dt \quad - \quad \int A_0A_1sin(\phi_0 t)cos(\phi_1 t)dt+\right.$$
$$\left.\int B_0A_1cos(\phi_0 t)cos(\phi_1 t)dt \quad - \quad \int A_0B_1sin(\phi_0 t)sin(\phi_1 t)dt\right] \quad (8)$$

For integrating this we define: $D = \phi_0 - \phi_1$ and $S = \phi_0 + \phi_1$.

Most easily, solutions for these four integrals can—for example—be taken from some internet resource. We get:

1.

$$\int B_0B_1cos(\phi_0 t)sin(\phi_1 t)dt = \frac{B_0B_1}{2}\left[-\frac{cos(Dt)}{D} - \frac{cos(St)}{S}\right] \quad (9)$$

2.

$$\int A_0A_1sin(\phi_0 t)cos(\phi_1 t)dt = \frac{A_0A_1}{2}\left[-\frac{cos(Dt)}{D} - \frac{cos(St)}{S}\right] \quad (10)$$

3.

$$\int B_0A_1cos(\phi_0 t)cos(\phi_1 t)dt = \frac{B_0A_1}{2}\left[\frac{sin(Dt)}{D} + \frac{sin(St)}{S}\right] \quad (11)$$

4.

$$\int A_0B_1sin(\phi_0 t)sin(\phi_1 t)dt = \frac{A_0B_1}{2}\left[\frac{sin(Dt)}{D} - \frac{sin(St)}{S}\right] \quad (12)$$

This combines to:

$$w(t) = \frac{\mu\phi_0}{2}\left[(-B_0B_1 - A_0A_1)\frac{\cos(Dt)}{D} \quad + \quad (-B_0B_1 - A_0A_1)\frac{\cos(St)}{S} + \right.$$
$$\left. (+B_0A_1 + A_0B_1)\frac{\sin(Dt)}{D} \quad + \quad (+B_0A_1 - A_0B_1)\frac{\sin(St)}{S}\right] \tag{13}$$

Hence:

$$w(t) = \frac{\mu\phi_0}{2}\left[(-B_0B_1 - A_0A_1)\left(\frac{\cos(Dt)}{D} + \frac{\cos(St)}{S}\right) + \right.$$
$$\left. (B_0A_1 + A_0B_1)\frac{\sin(Dt)}{D} + (B_0A_1 - A_0B_1)\frac{\sin(St)}{S}\right] \tag{14}$$

The second case concerns $\phi_0 = \phi_1 := \phi$.
This simplifies the above four integrals to:

1.

$$\int B_0B_1\cos(\phi t)\sin(\phi t)dt = B_0B_1\frac{1}{2\phi}\sin^2(\phi t) \tag{15}$$

2. and the next one is the same, only with different coefficients:

$$\int A_0A_1\sin(\phi t)\cos(\phi t)dt = A_0A_1\frac{1}{2\phi}\sin^2(\phi t) \tag{16}$$

3.

$$\int B_0A_1\cos(\phi t)\cos(\phi t)dt = B_0A_1\left[\frac{t}{2} + \frac{1}{4\phi}\sin(2\phi t)\right] \tag{17}$$

4. and similarly

$$\int A_0B_1\sin(\phi t)\sin(\phi t)dt = A_0B_1\left[\frac{t}{2} - \frac{1}{4\phi}\sin(2\phi t)\right] \tag{18}$$

First without collecting the first two coefficients we get:

$$w(t) = \mu\phi\left[B_0B_1\frac{1}{2\phi}\sin^2(\phi t) \quad - \quad A_0A_1\frac{1}{2\phi}\sin^2(\phi t) + \right.$$
$$\left. B_0A_1[\frac{t}{2} + \frac{1}{4\phi}\sin(2\phi t)] \quad - \quad A_0B_1[\frac{t}{2} - \frac{1}{4\phi}\sin(2\phi t)]\right] \tag{19}$$

and finally:

$$w(t) = \mu\phi\left[\frac{B_0B_1 - A_0A_1}{2\phi}\sin^2(\phi t) + \frac{(B_0A_1 - A_0B_1)t}{2} + \frac{B_0A_1 - A_0B_1}{4\phi}\sin(2\phi t)\right] \tag{20}$$

What is the rationale behind these calculations? If you have the Fourier spectra of any signals $u_0$ and $u_1$ then you will know all coefficients $A_0^0, \ldots, A_0^n$ as well as $A_1^0, \ldots, B_1^n$ because

$$u_j = A_j^0 + \sum_{i=1}^{n} \left[ A_j^i \cos(i\phi_j t) + B_j^i \sin(i\phi_j t) \right], \; for \; j = 0, 1 \qquad (21)$$

Then you can calculate the result analytically by using the full combinatorics $(A_0^i; B_0^i) \Leftrightarrow (A_1^k; B_1^k)$ with a complete index permutation over $i$ and $k$. Here the derivative of $u_0$ will here "turn the coefficients $A$, $B$ around" as cos+ sin becomes −sin+ cos. Note, however, that this holds only as long as the signals remain stable and their spectrum does not change. If there is only a slow drift, a windowing method might still be OK to update the weights with this method.

## Conclusion

In this study, we have demonstrated that it is possible to use differential Hebbian learning for efficient adaptive noise reduction. In a series of older papers, we had shown that ICO-learning reliably converges in closed-loop systems such that the agent always learns to successfully react to the earlier event [23, 24] and a later paper [25] showed how to transfer the simple one-neuron ICO-rule to a network implementation. Thus, the fact that the later signal $x_0$ is used as the error for learning leads to convergence. This is the case in those older studies, but also in the here-introduced system.

The here-used ICO learning rule is, in a stricter sense, not 'Hebbian' anymore, because it directly correlates inputs with each other. Hence, it is a heterosynaptic learning rule. These types of rules exist at neuronal dendrites where it has been demonstrated that heterosynaptic learning may play an important role [26–28].

The technical setup of this system is related to the LMS algorithm [10–13], but uses the temporal derivative in the learning rule. This is an advantageous concept, because the derivative is a predictor of the error signal due to its phase lead (see [7] for a discussion of this property). In addition to this, the rigorous convergence condition $x_0 = 0$ guarantees that learning stops with unchanging synaptic weights as soon as noise reduction has been successful Note that in technical systems one could introduce a threshold $\Theta$ and force $\hat{x}_0 = 0$ for $x_0 < \Theta$, setting $\hat{x}_0 = x_0$ otherwise. This should be done in case of small, remaining noise amplitudes where one would then use $\hat{x}_0$ for learning). Learning, however, will continue if signal complexity increases, and $x_0$ deviates from zero again, until the next stable weight configuration is reached (see e.g., Fig 4). In an earlier study, we could prove that this type of learning is equivalent to the learning of a controller that performs adaptive model-free feed-forward compensation [29], which is—in this application—a controller that eliminates the noise proactively by using the distant microphone signals.

This study, thus, has shown that it is possible to successfully transfer differential Hebbian learning, derived from the neurosciences, into a technical domain where we do not any longer rely on events (spikes) but can address time continuous signals, too.

## Acknowledgments

The authors are grateful for discussions with the European ADOPD consortium (www.adopd. eu) concerning the contents of this publication.

## Author Contributions

**Conceptualization:** David Kappel, Minija Tamosiunaite, Christian Tetzlaff.

**Formal analysis:** David Kappel, Minija Tamosiunaite.

**Funding acquisition:** Christian Tetzlaff, Florentin Wörgötter.

**Investigation:** Konstantin Möller, David Kappel, Minija Tamosiunaite.

**Methodology:** David Kappel, Minija Tamosiunaite, Bernd Porr.

**Project administration:** Florentin Wörgötter.

**Software:** Konstantin Möller, Minija Tamosiunaite.

**Supervision:** Florentin Wörgötter.

**Validation:** Bernd Porr.

**Writing – original draft:** Bernd Porr.

**Writing – review & editing:** David Kappel, Christian Tetzlaff, Florentin Wörgötter.

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
