## [Decision Letter · Decision Letter 0]

18 Apr 2022

PONE-D-22-08630Differential Hebbian learning with time-continuous signals for active noise reductionPLOS ONE

Dear Dr. Wörgötter,

Thank you for submitting your manuscript to PLOS ONE. After careful consideration, we feel that it has merit but does not fully meet PLOS ONE’s publication criteria as it currently stands. Therefore, we invite you to submit a revised version of the manuscript that addresses the points raised during the review process.

We look forward to receiving your revised manuscript.

Kind regards,

Felix Albu, Ph.D.

Academic Editor

PLOS ONE

Journal Requirements:

"We gratefully acknowledge funding from the European Commission under H2020 grant agreement 899265, FET-Open Project ”ADOPD”"

"F.W. and C.T. received funding from the European Commission under H2020 grant

agreement 899265, FET-Open Project "ADOPD".

https://ec.europa.eu/info/research-and-innovation/funding/funding-opportunities/funding-programmes-and-open-calls/horizon-2020_en

Additional Editor Comments:

The decision is Minor revision.

Reviewers' comments:

Reviewer's Responses to Questions

**Comments to the Author**

1. Is the manuscript technically sound, and do the data support the conclusions?

Reviewer #1: Yes

Reviewer #2: Yes

2. Has the statistical analysis been performed appropriately and rigorously? 

Reviewer #1: Yes

Reviewer #2: I Don't Know

3. Have the authors made all data underlying the findings in their manuscript fully available?

Reviewer #1: Yes

Reviewer #2: Yes

4. Is the manuscript presented in an intelligible fashion and written in standard English?

Reviewer #1: Yes

Reviewer #2: Yes

5. Review Comments to the Author

Reviewer #1: The main contribution of this paper is the show that it is possible to use a learning rule derived from spike timing dependent plasticity found in biological synapses for active noise active noise reduction. The proposed Differential Hebbian learning rule successfully reduces the noise under a range of different simulated configurations. In addition to pure noise cancellation, the method also allows for specific parts of an incoming noisy signal to pass through while filtering out the noise.

The paper show cases the practical potential of a bio-inspired learning rule that quickly adapts to changing noise reduction needs. The variety of different configurations used to test the methods highly strengthens its credibility for potential applications.

The authors provide thorough implementation details to aid replicabillity of the study.

Minor Suggestions:

- In the Results & Discussion section, a specifics of the series of experiments are introduced. The paper might have been a bit easier to read had the different configurations been listed in a section called Experiments (or similar). Then the Results & Discussion section could be reserved to presenting and discussing the results of the experiments.

- In the Conclusion, the authors refer to a series of older papers (line 228, line 428). Perhaps it would make sense to include these references in the introduction instead, to give the reader more context early on.

- In line 22, I think “had” should be replaced with “have”.

- Sentence starting in line 29 could benefit from a reformulation, perhaps with a deletion of “In general,”.

- The sentence starting in line 231: “Thus, the fact that the later signal x0 is used as the error for learning leads in those older studies, but also in the here-introduced system, to convergence.” is a bit hard to make sense of with the interposed sentences, and could perhaps be reformulated.

- Perhaps I don’t know the terminology around active noise reduction well enough, but it is not entirely clear to me what is meant by “at a high level” in sentences in line 135 and line 156.

Reviewer #2: The abbreviations ICO and ISO are not defined. A section on intuition regarding what ICO is doing and how it counteracts the noise would be useful. A little more detail in the learning rule section and noise reduction mechanism section would help the reader follow the arguments more easily.

6. PLOS authors have the option to publish the peer review history of their article (what does this mean?). If published, this will include your full peer review and any attached files.

Reviewer #1: No

Reviewer #2: No

---

## [Author Response · Author response to Decision Letter 0]

25 Apr 2022

Reply to the reviewers' comments is provided in the Response to Reviewers document.

---

## [Decision Letter · Decision Letter 1]

10 May 2022

Differential Hebbian learning with time-continuous signals for active noise reduction

PONE-D-22-08630R1

Dear Dr. Wörgötter,

We’re pleased to inform you that your manuscript has been judged scientifically suitable for publication and will be formally accepted for publication once it meets all outstanding technical requirements.

Kind regards,

Felix Albu, Ph.D.

Academic Editor

PLOS ONE

Additional Editor Comments (optional):

The decision is Accept.

Reviewers' comments:

Reviewer's Responses to Questions

**Comments to the Author**

1. If the authors have adequately addressed your comments raised in a previous round of review and you feel that this manuscript is now acceptable for publication, you may indicate that here to bypass the “Comments to the Author” section, enter your conflict of interest statement in the “Confidential to Editor” section, and submit your "Accept" recommendation.

Reviewer #1: All comments have been addressed

Reviewer #2: All comments have been addressed

2. Is the manuscript technically sound, and do the data support the conclusions?

Reviewer #1: Yes

Reviewer #2: Yes

3. Has the statistical analysis been performed appropriately and rigorously? 

Reviewer #1: Yes

Reviewer #2: Yes

4. Have the authors made all data underlying the findings in their manuscript fully available?

Reviewer #1: Yes

Reviewer #2: Yes

5. Is the manuscript presented in an intelligible fashion and written in standard English?

Reviewer #1: Yes

Reviewer #2: Yes

6. Review Comments to the Author

Reviewer #1: (No Response)

Reviewer #2: (No Response)

7. PLOS authors have the option to publish the peer review history of their article (what does this mean?). If published, this will include your full peer review and any attached files.

Reviewer #1: No

Reviewer #2: No

---

## [Editor Report · Acceptance letter]

12 May 2022

PONE-D-22-08630R1 

Differential Hebbian learning with time-continuous signals for active noise reduction 

Dear Dr. Wörgötter:

I'm pleased to inform you that your manuscript has been deemed suitable for publication in PLOS ONE. Congratulations! Your manuscript is now with our production department. 

Kind regards, 

on behalf of

Dr. Felix Albu 

Academic Editor

PLOS ONE